# Contribution of health system governance in delivering primary health care services for universal health coverage: A scoping review

**Resham B Khatri**[1,2]*, **Aklilu Endalamaw**[2,3], **Daniel Erku**[4,5], **Eskinder Wolka**[6],
**Frehiwot Nigatu**[6], **Anteneh Zewdie**[6], **Yibeltal Assefa**[2]

**1** Health Social Science and Development Research Institute, Kathmandu, Nepal, **2** School of Public Health, The University of Queensland, Brisbane, Australia, **3** College of Medicine and Health Sciences, Bahir Dar University, Bahir Dar, Ethiopia, **4** Menzies Health Institute Queensland, Griffith University, Brisbane, Australia, **5** Centre for Applied Health Economics, School of Medicine, Griffith University, Brisbane, Australia, **6** International Institute for Primary Health Care-Ethiopia, Addis Ababa, Ethiopia

\* rkchettri@gmail.com

## Abstract

### Background

The implementation of the primary health care (PHC) approach requires essential health system inputs, including structures, policies, programs, organization, and governance. Effective health system governance (HSG) is crucial in PHC systems and services, as it can significantly influence health service delivery. Therefore, understanding HSG in the context of PHC is vital for designing and implementing health programs that contribute to universal health coverage (UHC). This scoping review explores how health system governance contributes to delivering PHC services aimed at achieving UHC.

### Methods

We conducted a scoping review of published evidence on HSG in the delivery of PHC services toward UHC. Our search strategy focused on three key concepts: health system governance, PHC, and UHC. We followed Arksey and O'Malley's scoping review framework and adhered to the Preferred Reporting Items for Systematic Reviews and Meta-Analyses extension for Scoping Reviews (PRISMA-ScR) checklist to guide our methodology. We used the World Health Organization's framework on HSG to organize the data and present the findings.

### Results

Seventy-four studies were included in the final review. Various functions of HSG influenced PHC systems and services, including:1) formulating health policies and strategic plans (e.g., addressing epidemiological and demographic shifts and strategic financial planning), 2) implementing policy levers and tools (such as decentralization, regulation, workforce capacity, and supply chain management), 3) generating intelligence and evidence (including priority setting, monitoring, benchmarking, and evidence-informed decision-making),

**Data availability statement:** All relevant data are within the manuscript and its Supporting Information files.

**Funding:** The author(s) received no specific funding for this work.

**Competing interests:** The authors have declared that no competing interests exist.

4) ensuring accountability (through commitments to transparency), and 5) fostering coordination and collaboration (via subnational coordination, civil society engagement, and multisectoral partnerships). The complex interplay of these HSG interventions operates through intricate mechanisms, and has synergistic effects on PHC service delivery.

## Conclusion

PHC service delivery is closely linked to HSG functions, which include formulating strategic policies and plans responsive to evolving epidemiological and demographic needs, utilizing digital tools, decentralizing resources, and fostering multisectoral actions. Effective policy implementation requires robust regulation, evidence-based decision-making, and continuous monitoring. Accountability within health systems, alongside community engagement and civil society collaboration, is vital for realizing PHC principles. Local health institutions should collaborate with communities—end users of these systems—to implement formal rules and ensure PHC service delivery progresses toward UHC. Sociocultural contexts and community values should inform decision-making aligning health needs and services to achieve universal access to PHC services.

## Introduction

Health system governance (HSG) refers to the rules (formal and informal) for collective action and decision-making in a system with diverse players and organizations. In contrast, no formal control mechanism can dictate the relationship between those players and organizations [1,2]. Effective HSG ensures a strategic policy framework exists and is combined with effective oversight, coalition building, regulation, system design, and accountability in the health sector [3]. HSG acts as the maker and breaker of any health system, involving the degree of decision-making autonomy or discretion given to the managers and the rules constraining them [3,4]. The HSG emphasizes oversight and accountability arrangements, incentivizes organizations and their managers to deliver the mandates, and provides checks and balances [3].

The World Health Organization's (WHO) HSG framework has outlined five functions: formulating policy and strategic plans, generating intelligence, collaboration and coalition, and ensuring accountability [3]. The PHC measurement framework highlights people-centred care, functional mechanisms (supply and demand), and effective service delivery (e.g., community engagement, facility management and accessible, comprehensive healthcare) [5].

Governance and leadership (or HSG) are cross-cutting elements across all health system building blocks of health systems (e.g., health workforce, financing) to achieve goals (e.g., equity, efficiency, responsiveness). For instance, HSG functions deployed in the Asia-Pacific region to facilitate progress toward UHC included political commitment, leadership and support from politicians and civil servants, good stakeholder engagement, regulatory, political, and institutional structures to support policy implementation, and systems monitoring and evaluation [6]. HSG defines how other building blocks interact and focuses an internal organizational working environment conducive to improved health service delivery [7,8].

Health system governance differs in high- income countries (HICs) and low and middle-income countries (LMICs). For example, PHC services in high-income countries (HICs) are delivered by general practices, primarily primary care or family medicine [9]. The health system of HICs places a high priority on the disease burden of non-communicable diseases through research and innovation, technology and quality of care, alongside established health insurance programs and regulated health systems [10]. In contrast, health systems

in LMICs- including HSG- focus on improving access and coverage of health services (e.g., combating communicable and infectious diseases, and conditions of maternal, child health and nutrition), and service delivery by primary care workers (e.g., community health workers) at peripheral facilities and communities [10,11]. HSG in LMICs often grapples with workforce management and supervision, poor health insurance systems, weak regulation, inadequate digital governance, limited accountability and social responsibility, and poor quality of care [12].

Sturmberg and Martin argue that a clearer link exists between UHC, PHC and HSG [13]. The UHC centers on equitable financing, PHC approach focuses on appropriate and timely care, and HSG enables PHC implementation by involving local resources and communities [13]. The PHC approach is dynamic, shaped by power relationships among funders and health stakeholders, necessitating effective HSG [14,15]. Recent global health policies stress strengthening health systems, reforming policies and programs, structures and inputs, and formulating evidence based rules to enhance performance [3,16]. Achieving the global target of leaving no one behind requires efficient, resilient health systems underpinned by effective leadership and governance.

The effective HSG- supported by medical technologies, medicines, and e-health records- is critical for PHC service delivery [12,17]. Nonetheless, many health systems face resource shortage (human, financial) and weak digital infrastructure and evidence based policy-making and enabling environment (e.g., rules and regulations) [18]. While literature often examines standalone building blocks of health system (e.g., workforce, commodities, financing, health information), these are deeply intertwined with governance/leadership or stewardship, which shapes PHC service delivery. Few studies analyse the linkage of HSG interventions operate with other building blocks to advance UHC [8,19,20]. This study addresses the question: 1) What (and how) are the contributions of HSG (successes and challenges) to the production and delivery of PHC services toward UHC? The findings of aim to inform health system policymakers in designing evidence-informed strategies for PHC services toward UHC.

## Methods

We conducted a scoping review of published evidence reporting HSG in the PHC setting following the Preferred Reporting Items for Systematic Reviews and Meta-Analyses extension for Scoping Reviews (PRISMA-ScR) checklist (Supplementary Information, Table S1) [21]. We followed the methodological framework of Arksey and O'Malley [22], which was further refined by other researchers [23,24]. This scoping review framework contrasts with the procedures followed in systematic reviews, making it more useful to policymakers, practitioners, and service users [22]. We outlined this scoping study based on our recent experiences reviewing the literature on PHC [25–27]. We conceptualized three concepts: HSG, PHC and UHC. These concepts helped to define search strategies. Our research team assumed the search concepts were broad to provide a breadth of issues to explore in the review. The search concept was further clarified by preliminary discussion among authors and agreement on the topic's scope, breadth, and significance.

### Data sources and search strategy

We searched six electronic databases (PubMed, Web of Science, Scopus, Cochrane Library, Embase, and PsycINFO) that described HSG interventions implemented in PHC systems. Multiple databases were used to capture important studies for the review. PsycINFO was also included to capture related behavioural science-related studies and HSG topic, including social science approaches and strategies [28,29]. The search approach was followed by

complementary searches, including citation searches of included studies in Google Scholar to locate further eligible articles not identified in the database searches. The first ten pages of Google Scholar were searched to determine if relevant studies could be identified. The keywords used in the search strategy were built on three key concepts and tailored key search terms: a) governance (leadership, governance, stewardship, accountability, management, coordination, collaboration, regulation, multisectoral*, intersectoral*); b) "primary health care"; and c) UHC ("universal health care", health services accessibility", "quality of health care", "safe health care", "health coverage", "health care coverage", "health service coverage", "universal coverage", "universal health coverage", UHC, "essential health coverage", "health insurance coverage", "financial risk protection", "financial hardship", "financial protection", efficiency, equity, responsiveness, effectiveness, performance). Boolean operators (AND, OR) and truncations (*) varied depending on the database [Supplementary Information, Table S2].

## Inclusion and exclusion criteria

We used a similar approach tailored for each. The search included articles published in English from the inception of each database up to 31 July 2023. No time- or country-related limitations were applied. We included all relevant studies (e.g., quantitative, qualitative, mixed methods, review) that dealt with the HSG issue in PHC systems. We excluded records such as newspaper articles, newsletters, news releases, memorandums, blog posts, social media, letters to the editor, and correspondence). We included all studies about HSG that described its linkage with PHC and/or UHC based on inclusion criteria. We included those studies that focused on the contribution (successes and challenges) of HSG to PHC implementation.

## Selection of studies

Data were managed using EndNote version 20 software. The screening was initially undertaken by the first author based on the title and abstract and assessed by the second and third authors. This was followed by full-text screening initially by the first author and assessed by the second author. Any disagreements were resolved by discussion. The selection of studies took an iterative, holistic approach consistent with the PRISMA-ScR checklist [21]. We considered a study relevant if the data contributed to generate evidence for the review that can answer our review question, rather than prioritizing the quality of the individual studies [30,31]. Studies addressing our research objective were included regardless of methodological quality [30,32].

## Data analysis and synthesis

A data-charting form was developed to extract data from each study that covered author, year, country, type of study, key concepts, and main findings (Supplementary Information, Table S3). A descriptive-analytical method was used to extract contextual or process-oriented information. Data were extracted by the first author and double-checked by the second and third authors. Any disagreements on the extracted were discussed among authors' team and were resolved by discussion among authors. Thematic analysis of data was conducted by adopting Gale's framework method [33], involving steps such as collection of raw data, familiarisation with data, paraphrasing of data/labelling, developing/applying the analytical framework, matrix charting, and finally interpretation. After reading and familiarising the data, we extracted important concepts/categories. We grouped data extracts (or which contains similar ideas from different papers) into the WHO framework on governance and financing to synthesise and explain review findings [3], which includes five components/categories: a) Formulating policy and strategic plans; b) Generating intelligence; c) Putting in place levers

or tools for implementing policy); d) Collaboration and coalition-building across sectors/partners; and e) Ensuring accountability by putting in place. Themes were synthesized under these categories/components.

## Results

Search strategy accessed a total of 3729 records from PubMed/Medline (n = 1,353), Cochrane (n = 34); Scopus (n = 240); EMBASE (n = 970), Web of Science (n = 946), PsycINFO (n = 165), and Google Scholar (n = 21). Using EndNote 20, we removed duplicate articles and obtained 1408 articles. After screening titles and abstracts, we excluded 2,223 articles after title and abstract screening and leaving 98 full-text articles were assessed for eligibility assessment. Following a full text review, we excluded 24 articles. Finally, 74 articles were included in the final review (Fig 1).

### Descriptive characteristics of included studies

Of these 74 studies, 17 were from HICs (Europe (n = 1), Chile (n = 1), Estonia (n = 1), Sweden (n = 1), UK (n = 1), USA (n = 1), Canada (n = 1), Denmark (n = 1), New Zealand (n = 1), Armenia (n = 1), Seychelles (n = 1), Australia (n = 6)) [Supplementary Information, Table S4]. Meanwhile, 12 studies were from upper-middle-income countries (Mexico (n = 1), Cuba (n = 1),

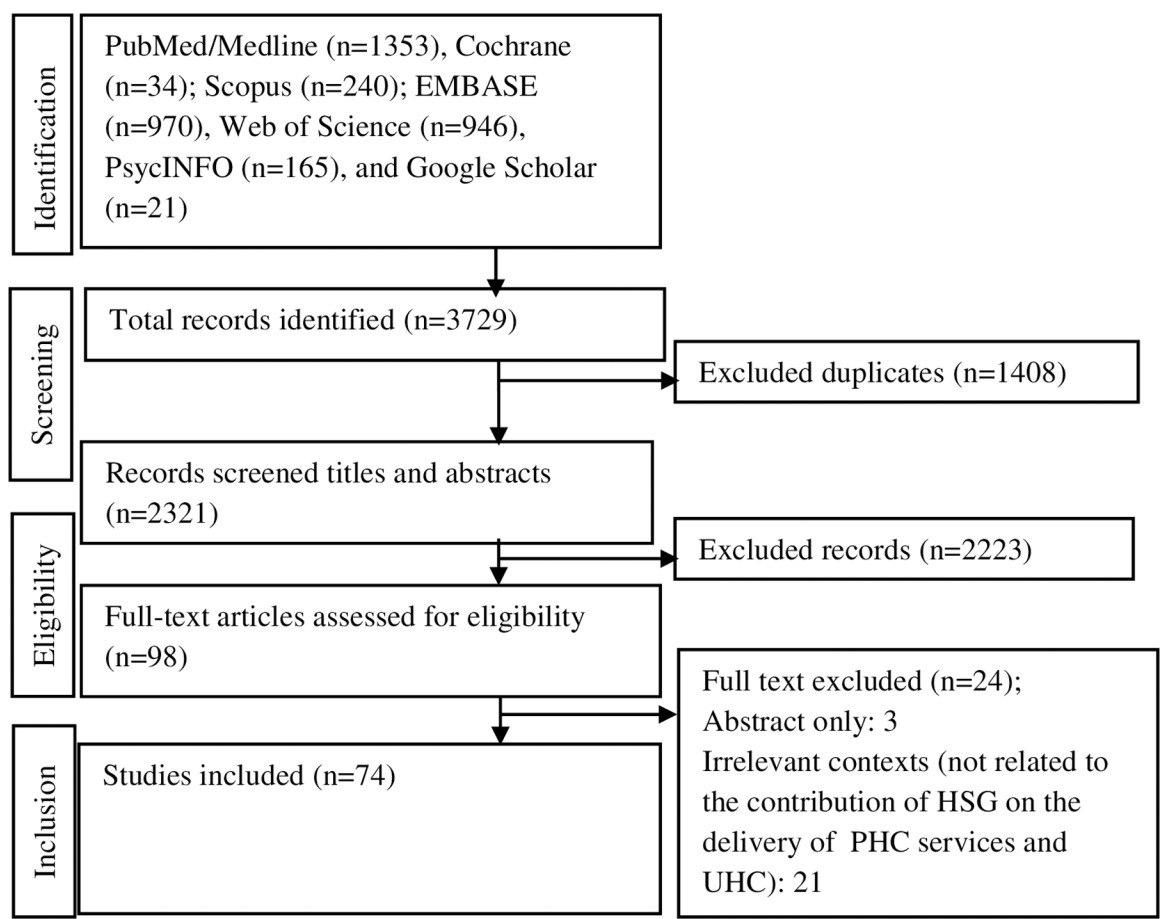

**Fig 1. Flow chart showing the selection of studies for this study.**

Saudi Arabia (n = 1), Ukraine (n = 1), El Salvador (n = 1), Malaysia (n=), Indonesia (n = 1), South Africa (n = 2), China (n = 3)). Additionally, 28 studies from low- and lower-middle-income countries (Namibia (n = 1), Ghana (n = 1), Zambia (n = 1), Egypt (n = 1), Kenya (n = 1), Liberia (n = 1), Mozambique (n = 1), Uganda (n = 2), India (n = 2), Tanzania (n = 2), Nigeria (n = 3) and Iran (n = 3), Ethiopia (n = 4), and Nepal (n = 5)). Seventeen studies were conducted across multiple low and lower-income countries from different regions. The included studies encompassed various study designs, including reviews (n = 16), quantitative studies (n = 10), qualitative studies (n = 28), mixed methods (n = 10), policy analysis and discussion papers (n = 10) [Supplementary Information, Table S4].

**Formulating policy and strategic plans.** Studies reported multiple strategic health plans and financial governance issues to address changing demographic and epidemiological shifts and socioeconomic and health system contexts.

**Health policy and plans:** Leadership in the health system is intricately linked with other building blocks, creating the enabling environment in the health sector, and providing national laws and policies for planning and budgeting health services [34,35]. Legal policy framework (laws and regulations), national financing policies and plans, and the capacity of subnational governments have improved the implementation of PHC [36–38]. For instance, the Estonian health system reform (which included changes in laws, restructuring of organizations) and Iranian health reform (which focussed on reforms in management, leadership, and human resource development) ensured coordination between policy and operational levels [39,40]. Moreover, Denmark and Sweden invested in collective healthcare systems for efficient PHC service delivery toward UHC [41,42].

Leadership with political power can take bold steps in resource allocations, setting policy criteria to ensure infrastructure development and a robust health workforce [43–45]. Additionally, leadership capacity in health (such as relationships with supporting organizations, higher-level roles in policymaking, stability, and supportive policy) was essential for financial management and performance improvement to achieve UHC in PHC services [45–48].

Some countries focus on policy and plans for changing the context of epidemiology and demography. In China, the focus was on strengthening infrastructure for improved service towards equity and universality of health services [49]. Seychelles' adapted health systems, public expectations, and demand for high-quality health services in changing epidemiological and demographic needs [50]. In Iran, reforms and investment in PHC systems ensured human resource development to rehabilitate ageing infrastructures [51]. The availability and integration of mental health policy into other health policies and the presence of non-governmental organizations (NGOs) were priorities for mental health services [52].

However, inadequate political and technical leadership, weak resource and information management, and piecemeal plans have hindered countries from realizing UHC [36,37,53,54]. Many countries lacked policy leadership on Non-Communicable Diseases (NCDs), resulting in market-orientated solutions for NCDs [36,41,55]. In Nigeria, dysfunctions of the PHC system hindered health services' sustainable and equitable provision [56]. Thus, addressing issues of HSG requires changes in legal and procedure documents (e.g., rules, and regulations) that influence resource allocations, which often impacts other health system building blocks [13,47].

**Financial governance and planning:** Studies reported improvement in financial schemes to prevent and control diseases, such as community health financing, equity-informed financing models for health insurance, prepaid financing, government subsidies in domestic financing and social insurance schemes [57–60]. These financial schemes have made substantial progress in financial protection and reduced high out-of-pocket payment (OOP) expenses, thereby improving access to care [49,61]. Additionally, the Estonian health system invested in

preventing and managing chronic conditions in PHC settings, which reduced hospital admission rates [39].

Modifying provider payment systems has led to service innovations and human resource (e.g., financial viability for general practitioners), which influenced the availability of after-hours services and improved PHC performance in service delivery [39,62,63]. Furthermore, financial incentives in PHC programs and developing funding sources improved the equitable and efficient delivery of integrated health services [14,42,61,64]. User fees remain a predominant revenue source, and urban facilities have prioritized expenditures toward drugs and supplies [65]. Health insurance programs have had positive effects, but discontinuing performance-based incentives to providers has harmed the quality of maternal health services [66,67].

Many health systems face health inequities due to poor financial planning and governance in the health sector. Inadequate financing and cash flow, lack of expansion of social health insurance, and regressive subnational expenditure pose challenges for good financial management [38,61,68–70]. Moreover, financing and governance are linked to the root causes of the dysfunctions in the PHC system, which led to inequitable service coverage and high OOP in health care [56,60]. Additionally, stagnating economic growth in LMICs and reliance on external funding have resulted in inadequate preventive services for NCDs [52,61,62].

The emergent priority of resource allocation and expenditure control has further marginalized the health service in community health centres [41,49,71]. Many PHC systems are underdeveloped for NCDs (e.g., mental health), and the focus has been on hospital-based care in cities. Such systems lack district-level planning, adequate record-keeping systems, and prioritization of referral system [36,52,55]. A slow transition of service provision, fragmented health services, suboptimal quality of care, inefficiency, and poor public satisfaction have further constrained NCD control and prevention prioritization [34,37,41,49,56,60,69]. Poor financial access (e.g., insufficient funding support) and increased costs have compounded inequitable access to and inadequate referral for health services [36,47,69,71].

**Implementing policy levers and tools.** Decentralization, digitalization, and regulation serve as policy levers and tools in PHC systems. These levers are essential for mobilizing resources (human and logistics) to implement and deliver PHC services.

**Decentralization, regulation, and digitalization:** Decentralization in the health sector is vital for reforming leadership and program implementation. Communities stakeholders have identified for decentralisation: increasing the role of communities and service users in governmental bodies; establishing local coordination/working groups; developing community-based spaces for integrated service provision; embedding programs in the existing services; advocacy and lobbying leaders and service users; increasing capacity of communities in financial management [72]. Iran's decentralized health system improved customer orientation and the performance of budgeting system, empowering community members and local stakeholders to exercise political rights and raise questions [51,73]. In Ukraine, decentralisation led to the allocation of funds alongside responsibilities for developing community services [72].

Stewardship capacity depends on regulation, while regulation depends on decentralization [14,42,74]. Government-dominated centralized decision-making has reinforced skewed resource allocation and prioritization, leading to health inequities [46,75]. However, decentralization has sometimes resulted in fragmentation of the PHC program due to leadership, coordination, and collaboration problems; infrastructure, physical accessibility, and financial challenges; PHC workforce shortages and lack of competencies; low awareness of available services and high stigma; and issues related to war, crises, and pandemics [62,72].

Cuba's national regulations encouraged the involvement of social sectors in health policies and programs [76]. The responsibilities of subnational governments include establishing

financial regulations to align their programs with national policies [38]. Funding regulation (gatekeeping) and commitment toward transparency have ensured accountability and improved PHC performance and quality service delivery [14,61,62,77]. Regulatory decision support systems, accreditation, and flexible organizational culture are determinants of health system performance [42,51]. Furthermore, regulation is fundamental in implementing policies (e.g., mental health policy) to protect the rights of people with mental illness [52].

Regulation and use of digital tools are necessary policy levers. Regulated private sector engagement in vouchers and contracting programs has expanded health services in the Asia Pacific region [74]. The challenges of private health insurance schemes include the lack of regulation, the exclusion of informal workers in weak community engagement, and fragmented health services (public versus private) [62]. Moreover, innovations in information communication technology (e.g., telemedicine, WhatsApp-supported team consultation) have improved the quality and accessibility of services in many settings [48,64,78,79]. However, poor interoperability of technology and information systems, limited capacity, and poor supply chain management have hindered the PHC programs [34,37,47,56,60,64,69].

**Workforce skills and commodities:** Health system building blocks like human resources (HR) are essential for PHC service delivery. The HSG is intricately linked with visible inputs of health systems (e.g., HR, infrastructure) and invisible but strategic issues (e.g., relational aspects, norms, values) [58,80]. Providing skilled and trained staff, ensuring the availability of female health workers, task shifting, skill mix HR, providers' competence, confidence, and coordinating community actors are crucial in navigating the complexities of health community health systems [37,48,73,79]. Redistribution of a mix of trained providers (e.g., appropriate knowledge and personal skills) and their performance management have improved quality services to poorer communities [43,44,75,81]. Evidence suggests an interdisciplinary PHC team (including health promoters) bring holistic health care closer to the communities in El Salvador [82]. Inventory management and clinical performance of staff promote uninterrupted availability of supplies and equipment, thereby improving facility performance [83,84].

Difficulties in arranging medical products, e.g., drugs, equipment, vaccines) constrain policy implementation. Challenges include segmented supply chains, lack of communication information systems, limited technical capacity, poor infrastructure, shortages of the workforce, and inadequate supervision [37,47,56,60,64,69]. Furthermore, these inputs and process factors have further hindered services provision and delivery of PHC, including NCDs (e.g., mental health) [52,62,82].

**Generating intelligence  Evidence-based planning and monitoring:** Some countries have adopted evidence-informed decision-making and planning, monitoring policies and programs and generating insights for governance. Data analysis supports monitoring comprehensive national health plans, health-related services, and facilities [55,85]. Quality evidence and information inform decision-making in health planning and enhance service tracking and monitoring of service coverage [78]. For instance, quality services and continuity of care for chronic patients require access to quality information and health planning [43,82]. Studies revealed that countries have adopted evidence-informed reforms in Iran (evaluation and auditing of health management information system) [40], Mexico (use of evidence intensified) [35], the Western Pacific Region (evidence-informed priority setting for health benefits packages) [61], and Liberia's CHWs program (evidence-based policy and planning) [86].

Generating intelligence from information requires the processing of data and knowledge generation. However, challenges exist in generating evidence from information due to limited finances, lack of technical capacity at the district level, and lack of information systems [34,36,47,54,69]. Moreover, critical knowledge gaps exist to ensure context-specific

governance (e.g., governance of financing, workforce, health policymaking process accountability mechanisms) in public health emergencies [36,54,58].

**Ensuring accountability  Commitment towards accountability:**    Commitment, transparency, and participation are key features of accountability mechanisms that are important for health financing. Accountability with transparency ensure public participation and inclusion of public voices [50,85,87]. The public commitment towards accountability and transparency in policy programs has increased the responsibility for improving performance to ensure the equitable quality of health services [61,77,85,87]. Seychelles' high political commitment to invest in PHC and downward accountability has succeeded in achieving health system goals [50].

Institutionalizing local policies, community engagement, and cross-cutting priorities of funding and commitment have influenced transparency, accountability, and health system performance [57,75,77]. Additionally, a shift in capital expenditure allocation targeting poor communities has contributed to achieving health system objectives [75,77].

Civil societies and citizen groups have activated political or formal bureaucratic accountability channels which further negotiated national and local-level mandates to respond to socio-political and health system contexts [2,88]. Relationships between providers and communities, public reporting systems, and local organisations' performance improvement can drive broader population health goals [2,88,89]. In EI Salvador, social controllership and community engagement in local health systems ensured community accountability [82].

Poor accountability and weak transparency have influenced the decision-making process [73]. Additionally, meeting the expectations of powerful managers and responding to the expectations of citizens and patients are constrained by external and bureaucratic accountability mechanisms [90]. Current health systems experience a lack of accountability framework (e.g., funding, knowledge gaps) that could marginalize the implementation of PHC at the local level [58,71].

**Collaboration and coordination.**  Coordination and collaboration at the subnational level, with civil society organizations and intersectoral coordination, are effective in governing PHC systems.

**Subnational coordination:**    Operationalizing and implementing national mandates (laws, regulations, policies, and standards) require subnational organizational leadership, responsibilities, and capacity [38,44]. The leadership capacity of sub-national governments ensures the implementation of centrally defined PHC programs at the local level [38,78,90]. Moreover, public health legislation, community engagement, and collaborative planning depend on local governments' responsiveness and capacity [70,91].

Managerial skills and institutional capacity are vital for developing and implementing subnational-level integrated programs and plans [70]. The district health system is a unit that organizes strategic plans, manages resources, and engages stakeholders in national implementation programs [92]. In Australia, regional organizations (e.g., the local hospital and public health networks) sought control of the policy of steering market-based professionals, which resulted in the acceptability, appropriateness, and affordability of PHC services [63,89]. Adequate management capacity, engagement of local leadership, and active community accountability are other factors of PHC governance [84].

Studies reported that leadership management and governance, such as facility heads, strengthened management competencies for improved district capacity, structure and management practices and quality of health services [79,93]. Community-based health planning and services are public health risk protection strategies [43]. System inputs (joint planning, incentives and continuing education, supervision, change management, and better alignment of the responsibilities) effectively implemented centrally defined policies and

programs [38,78,90]. However, such top-down targets weakened district-level capacity in the context of local governments' limited resources (time and financial support, capacity). Additionally, they influenced the collaboration with the priority population and their needs [34,90,91].

**Civil society engagement:**    Community participation can improve the performance of PHC systems. The involvement of multilevel stakeholders (users, providers, and government authorities) in health committees and community health organizations helped understand, interact, and translate policy vision within local spaces [62,94]. Community engagement and local leaders' support can enhance the management of local facilities for resource mobilization [81,95]. In Karnataka, horizontal coordination of local bodies demanded accountability [94]. Aboriginal community-controlled health organizations strengthened community capacity for the broader health system to implement comprehensive PHC targeting indigenous populations in Australia [96].

Stakeholder forums, committees, and networks of community health volunteers and retired government workers can have decisive influence and political connections to initiate formal mechanisms for strengthening community-based PHC units [59,81,95]. Reaching out to the community and lobbying governments for support (government support, operating existing sociocultural structures) was vital for participating disadvantaged groups (e.g., minorities and females) in health committees [97,98].

Participation of civil society organizations (e.g., NGOs) has been recognized as key partners in equitable and sustainable national development. Resource allocations, collective visions for resource mobilization and integrated planning for PHC services could meet local needs and increase the responsiveness of health systems [2,46]. Citizen groups have activated political or formal bureaucratic accountability channels, influencing provider responsiveness [88]. Community participation and ownership were vital within the socially constructed organizations of health system and their hierarchical functions to track the progress of integrated service delivery [13,40,47,57,64]. Civil society organizations (e.g., the Healthy Caribbean Coalition) can link policymakers and government authorities to ensure accountability for advancing health equity [58,87]. Health facilities with better management committees had better performance scores, especially in urban and private areas, leading to improved health facility delivery rates [99–101].

Moreover, civil society groups are watchdogs, resource brokers, partnership developers and advocates [73]. In PHC, prioritizing community engagement and reflecting on the diversity of community health ecosystem can address local needs through a whole-of-society approach [57,73]. In contrast, health providers' perception of the legitimacy of citizen groups and their support mediated the public's demands for better health services [88].

**Multisectoral engagement:**    Multisectoral policy and actions in health differentiate PHC from service-focused primary care by creating healthy living conditions and integrating cross-sectoral policy decisions. Multisectoralism moves from fragmentation to integration, institutionalization of activities, and mobilization of community resources, empowering people and communities, identifying local priorities of issues, and co-production of people's health, and engaging stakeholders in the whole-of-society effort [13,57,58]. Building resilient health systems requires multisectoral actions [102]. Community empowerment and control mechanisms are key to advancing PHC for improved health services coverage [60,82,87]. To achieve this, progressive health system reform is needed for community interactions to align with other actors in delivering comprehensive PHC [57,68].

The horizontal integration of the functions of health systems and community-based strategies can effectively design implementation strategies that favor changes to reorganize the system [2,35,81]. In addition, the institutionalization of policy and programs was instrumental

in improving PHC systems' performance in terms of responsiveness, efficiency, and effectiveness of health programs [59,70,86].

The integration of services and sectors (e.g., primary care and public health) and coordination of levels of care (prevention and health promotion for NCDs) are critical [13,75,82]. Regular outreach services and comprehensive PHC can meet increased health needs, defined service packages, a continuum of care, and appropriate referrals [35,61]. Comprehensive and people-and community-centred services require implementation of the PHC approach with high-level political commitment towards UHC, as seen in Cuba [76,87].

**Community engagement for delivery of PHC services:**    Local health committees' meetings functioned only when funds were available [95]. Power asymmetries within committees (e.g., service providers and users), token participation (e.g., disadvantaged groups), and the influence of powerful elites hindered decision-making and health system performance [97,98,103]. Increased social interactions and relationships among implementers enhanced communication and created opportunities for social learning, while cyclical performance monitoring and information flow contributed to system-wide effects [104]. Attributes of inadequate collaboration at the local level included a lack of verbal and formal complaint mechanisms (e.g., suggestion boxes), lack of responsiveness to providers or users' knowledge of entitlements or complaint mechanisms, lack of alternative providers, limited involvement of consumers and stakeholders, insufficient coordination, weak community network [37,103].

The availability of health workers and community health planning schemes filled gaps in geographical access and improved access to primary care, resulting in reduced healthcare costs [43,48,56]. In addition, the CHW's approach improved access to care, built trust, and increased the demand for and utilization of health services [68]. For example, in Liberia, an incentive-based community health assistants program adopted a systems approach, established coordination and partnership support, strengthened community engagement, and ensured PHC services for disadvantaged populations [86].

Poorly functioning collaborative community engagement and multisectoral actions resulted in inadequate resource mobilization and service delivery [95]. Donor-driven management and funding led to fragmentation of health services (disease-specific training) and multiple competing actors with little coordination [34,68,69,80]. Insufficient policymaker-implementer interactions hindering achieving PHC goals in LMICs and undermined the realization of UHC [37,53].

## Discussion

This review revealed several HSG interventions supporting the implementation of PHC towards UHC. Evidence-informed policies and plans to address the changing health needs of the community, financial governance, decentralization, regulation, digitalization, skills of the health workforce, use of community-level data, providers' responsiveness to context, public reporting, community engagement, subnational capacity, and civil society and multisectoral collaboration were key successes of good HSG in the PHC context. These factors can be considered strategic and operational levers for the effective implementation of PHC.

Formulating strategic policies and plans is vital to address the changing health needs through prevention-focused, equity-informed financial planning. This review suggests that using evidence and information to inform decision-making and planning, decentralizing resources, and information generation and management through technology and digital tools are critical. Digitalization in health systems has the potential for evidence generation and monitoring in policy and strategic planning; however, digital interoperability is inadequate in many LMICs. Previous evidence also revealed that investment in digital systems is essential for information management and knowledge generation within health programs, services, and

fiscal management [105]. Using digital tools and evidence can effectively inform policies and plans, track policy implementation, ensure an accountability system at all levels and reduce corruption in the health sector [105–107]. In LMICs, inadequate financing emerged as the biggest challenge for good HSG to ensure strategic policy development, technical capacity at sub-national levels to develop and implement integrated plans, and digital interoperability [70]. China's health sector reform regarding strategic policies and plans serve as an example where significant progress has been made in addressing inequities by reforming its healthcare delivery system (e.g., removing markups for drug sales, adjusting fee schedules, reforming provider payment, and enhancing financial protection for lower socioeconomic groups) [49].

Using policy levers and tools such as decentralization and regulation are effective in health leadership and governance. Previous reviews on decentralization in the health sector highlight increased equity, efficiency, and health system performance by bringing health services near the community [108]. Decentralization entails the delegation of authorities and resources, ensuring accountability and community engagement in executing these responsibilities. However, implementation might be hindered by limited capacity to allocate resources and minimize fragmentation [109]. Therefore, strengthening key policy levers is vital to promote health system responsiveness, efficiency, and effectiveness [70].

Empowered citizen groups and better accountability mechanisms are vital for health system governance. Downward accountability from providers with clear responsibilities was found to underpin health service delivery [88]. Furthermore, advocacy efforts and citizen groups can raise their voices for health rights and engage in collective actions to strengthen PHC system. Conversely, civil society, citizen groups, and community-based organizations can hold providers accountable, inform them of their duties and engage with local health systems [87]. Good HSG requires a public commitment to accountability, transparency, and participation, all of which are in the PHC context. Citizens' boards supporting the claiming rights and ensuring access to information are tools for improving accountability mechanisms. Providers' accountability to people (downward accountability) could enhance system efficiency.

Additionally, civil society accountability and the engagement of influential people (e.g., champions or networking with political leaders) on committee are vital for resource mobilization, local agenda setting and health service delivery [90]. In Nepal, a study revealed that social accountability interventions improved maternal health service quality by enhancing health system responsiveness, community ownership, tackling inequalities, and enabling communities to influence policy decisions [110]. Poor accountability and lack of commitment to shared objectives hinder policy implementation, particularly in financing and regulation under decentralized systems [14]. Thus, strengthening downward accountability (providers to communities), upward accountability (within hierarchical systems), and horizontal accountability (across stakeholders) is critical [111].

Collaboration and coordination among all sectors, actors, and systems components—especially subnational entities, civil society, and communities— are essential to produce and deliver health service with community acceptance. Collaboration between health care, social services, and other sectors is widely promoted as a pathway to improved health outcomes [112]. Community and civil society can contribute to the PHC systems by acting as watchdogs and gatekeepers of local health systems. At the local level, institutional arrangements (e.g., health facility committees, civil society engagement, and periodic meetings) enhance collaboration and foster horizontal coordination among stakeholders to implement health programs [113]. At the higher level, secretariats, working groups, and intersectoral cluster meetings provide strategic directions and guidance of the PHC policy approach. Developing workable mechanisms for multisectoral collaboration is crucial [70]. The HSG in PHC settings has

grown complex, involving multiple partners and requiring diverse interventions models, and approaches for transparency, accountability, participation, integrity, and capacity. Thus, systematic and multisectoral governance components must be prioritized to ensure design and implementation of evidenced based health policies [114].

Global health policies and strategies prioritize on ensuring health services to people already left behind, socially and geographically. For instance, Sustainable Development Goal 3 aims to achieve universal coverage of quality essential health services by 2030. Implementing the PHC approach is central to achieving UHC. Effective HSG supports the application PHC principles. Since health systems do not operate in isolation, effective HSG is vital, particularly in PHC systems within countries with fragile health systems. For effective HSG, governments must focus on both visible components (e.g., health infrastructure, health workforce, digital infrastructure) and less visible but strategic components/interventions (e.g., strategic plans, decentralization, moral values/accountability, coordination). Health system reforms targeting tactical and strategic components have reduced health inequities toward UHC (e.g., Turkey) [115]. Furthermore, the macro-health system also influences HSG and the implementation of PHC at the local level. Therefore, leaders of health systems must strengthen all health system building blocks across levels.

Findings suggest that health system performance in delivering PHC services can improve through financial incentives, regulation, community engagement, and quality enhancements. Studies from the LMICs highlight improvements in accountability, social responsibility, public-private partnership, provider-user communication, financial incentives, regulation, quality improvement, and community involvement [5,12,62]. Challenges in LMICs include inadequate service coverage, inequitable access, slow transitions to NCD-focused care, poor quality, and high OOP expenditures [37,60,94]. In HICs, policymakers focus on governance improvements, quality enhancement, and information technology [77,116]. Better measurement and dissemination of effective models are fundamental to advancing PHC delivery and outcomes in LMICs [5]. Decentralizing services to grassroots levels, stakeholder support, equitable resource distribution, organizational accreditation, and integrating quality/equity indicators into monitoring systems are essential for UHC-oriented programs [4]. Decision-making authority, coordination, resource control, development initiatives, and management skills drive health systems reforms PHC delivery towards UHC [117]. PHC implementation requires policies, strategies and programmes aligned with national priorities [60]. The beyond building block framework underscores governance's central role in mobilizing and institutionalizing resources for PHC delivery towards UHC [118]. Strengthening resilient health systems via infrastructure, skilled workforces, and financial risk protection is essential to address epidemiological and demographic challenges [119,120].

**Study strengths and limitations.** This scoping review synthesized findings from diverse studies, offering comprehensive insights. However, it lacked a registered or published protocol, and did not assess evidence quality. Other limitations include the exclusion of non-English studies and the absence of expert consultation to triangulate findings. Future research should integrate stakeholder consultations to validate results.

## Conclusion

Implementing the PHC approach relies on effective HSG, encompassing strategic policies and plans to address changing health needs, leveraging digital tools, decentralizing resources, and fostering multisectoral collaboration. Successful implementation of HSG requires robust regulation, evidence-based decision-making, and continuous monitoring. Additionally, accountability from health systems, communities, civil society, is essential to uphold PHC principles. Local health institutions bear the primary responsibility for engaging with communities—the

end users of these systems—to implement formal rules and ensure PHC service delivery progresses toward UHC. Sociocultural contexts and community values should inform decision-making aligning health needs and services to achieve universal access to PHC services.

## Supplementary information

**S1 Table. Preferred Reporting Items for Systematic Reviews and Meta-Analyses Extension for Scoping Reviews (PRISMA-ScR) Checklist.**
(DOCX)

**S2 Table. Tailored Search Terms for Different Databases.**
(DOCX)

**S3 Table. Data Extraction on Health System Governance (HSG) in Primary Health Care (PHC) from Studies Included in the Review.**
(DOCX)

**S4 Table. Descriptive Summary of Studies Included in the Review .**
(DOCX)

## Acknowledgements

None

## Author contributions

**Conceptualization:** Resham B Khatri, Eskinder Wolka, Yibeltal Assefa.

**Data curation:** Resham B Khatri, Frehiwot Nigatu, Anteneh Zewdie.

**Formal analysis:** Resham B Khatri, Aklilu Endalamaw, Daniel Erku, Yibeltal Assefa.

**Investigation:** Resham B Khatri, Aklilu Endalamaw, Yibeltal Assefa.

**Methodology:** Resham B Khatri, Aklilu Endalamaw, Daniel Erku, Eskinder Wolka, Frehiwot Nigatu, Anteneh Zewdie, Yibeltal Assefa.

**Project administration:** Resham B Khatri, Yibeltal Assefa.

**Resources:** Eskinder Wolka, Frehiwot Nigatu, Anteneh Zewdie, Yibeltal Assefa.

**Software:** Resham B Khatri.

**Supervision:** Resham B Khatri, Yibeltal Assefa.

**Validation:** Resham B Khatri, Aklilu Endalamaw, Daniel Erku, Eskinder Wolka, Frehiwot Nigatu, Anteneh Zewdie, Yibeltal Assefa.

**Visualization:** Resham B Khatri, Aklilu Endalamaw, Yibeltal Assefa.

**Writing – original draft:** Resham B Khatri, Yibeltal Assefa.

**Writing – review & editing:** Resham B Khatri, Aklilu Endalamaw, Daniel Erku, Eskinder Wolka, Frehiwot Nigatu, Anteneh Zewdie.

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
