## [Decision Letter · Decision Letter 0]

21 Feb 2024

PONE-D-23-27336Governance of primary health care systems and services: a scoping reviewPLOS ONE

Dear Dr. Khatri,

Thank you for submitting your manuscript to PLOS ONE. After careful consideration, we feel that it has merit but does not fully meet PLOS ONE’s publication criteria as it currently stands. Therefore, we invite you to submit a revised version of the manuscript that addresses the points raised during the review process.

Kindly find attached my comments and as well as reviewer1.

Indicate which changes you require for acceptance versus which changes you recommendAddress any conflicts between the reviews so that it's clear which advice the authors should followProvide specific feedback from your evaluation of the manuscript

We look forward to receiving your revised manuscript.

Kind regards,

Desire Aime Nshimirimana, MBChB,Msc

Academic Editor

PLOS ONE

Journal Requirements:

Additional Editor Comments:

Title: The title needs to be reviewed. “Governance of primary health care system”. What is the gap in the primary health care system? Are you talking about the influence of governance? Are you talking about the impact of healthcare governance on primary health system? The title needs to be amended. The title and objective must be linked. “In the objective, you have “explores function” and “UHC”. The title should have UHC.

Kindly refer to your research question to formulate the title “Line 115” “We identified the research question focusing on the contribution of HSG to delivery of PHC services”

1. Introduction

1. The first paragraph of introduction is having some repetitions: Line52 and Line55:

Line52: Health system governance (HSG) is the rules (formal and informal) for collective action

Line55: Additionally, HSG is “as the collective actions…”

Line 59-63: This paragraph is not clear. Kindly break the paragraph in to pieces to make it easy to understand for the reader

There are inconsistencies between the aim given in abstract and the one given in the introduction.

Line25: In the abstract the aim of the study: “This scoping review explores the function of HSG in the context of PHC setting towards UHC”.

Line102: The aim of the study; “This study aimed to synthesize HSG interventions' contribution to delivering and utilizing PHC services”

Page64-82: this paragraph enumerates frameworks: when I read these frameworks, I would expect to see the frameworks utilized in the document but it is not the case.

Instead, kindly discuss at least 3 studies on health systems governance of primary healthcare in the introduction and let the 3 studies support the discussion and universal health coverage!

Introduction should also briefly discuss Health systems blocks because these must comeback in the discussion.

2. Methodology

This framework of Arksey and O'Malley should be briefly described

Line 116: We conceptualized three concepts: role of HSG to deliver health service to achieve UHC; Where are the 3 concepts????

Identifying relevant studies

1. What guided the choice of the 6 databases? I have a concern on why you have chosen “PsychInfo” as one of the databases to search from. PsychInfo is a database of abstracts and articles in the field of psychology and psychiatry. Why did you choose this database? This needs a strong justification.

2. For each database, provide the search strategy including the search terms. Provide also the search strategy for google scholar engine.

Study selection

It is not clear how you identified relevant studies. Kindly provide clearly inclusion and exclusion criteria

Summarizing and reporting

It is confusing how you summarized and reported the results. Kindly elaborate how you did a synthesis in details, how you grouped the papers according to the groups they belong

3. Results

PRISMA: “Excluded as did not match purpose of the review”. This statement is not simply enough, kindly elaborate for each paper the reason of exclusion.

Line183: “In the final review, we included 64 studies”, yet in the PRISMA, the number of papers are 74.

Descriptive characteristics of included studies

Kindly describe all papers identified by country, type of paper, author and year with citation of each paper

Move Table1 to supplementary documents

Line 202: Fig2 presents the summary of HSG interventions in the PHC context

Is this Fig2 part of your results? In which paper did you extract this fig2? If the information of figure2 is not among the results, kindly put it in the introduction or literature review

Line 164: You said that to summarize the findings you used WHO framework on governance and financing: Looking in to the summary, they are many subtitles which are not part of WHO framework such multi-sectoral engagement (line 438), community engagement (line 459),….

Line 570: Our purpose of the review was to synthesize evidence rather than grade the evidence, however, the results are a simple summary of what was reported in the studies, on the contrary, the paper should synthesize the data as opposed to summarize. Results should have a synthesis and not a summary.

In the aim of the study and methodology, Universal health coverage appeared both in aim and search but does not come out among findings. What happened in the findings? Is it that no studies reported universal coverage?

4. Discussion: Discuss how performing HSG (countries) are doing, what special strategies they have in terms of HSG which are making them performing and their strengths and weaknesses. Discuss countries in pools of LMICs and HICs, what are differences

Conclusion

The conclusion is not supporting the findings at all. Kindly relate your findings to the presented results

Reviewers' comments:

Reviewer's Responses to Questions

**Comments to the Author**

1. Is the manuscript technically sound, and do the data support the conclusions?

Reviewer #1: Yes

2. Has the statistical analysis been performed appropriately and rigorously? 

Reviewer #1: N/A

3. Have the authors made all data underlying the findings in their manuscript fully available?

Reviewer #1: Yes

4. Is the manuscript presented in an intelligible fashion and written in standard English?

Reviewer #1: No

5. Review Comments to the Author

Reviewer #1: The work seem to be done in a hurry and requires editing and revision of sentence constructions in most cases. There are some inconsistencies in the authors claim. In addition the author should structure themes in a way to guide the author to identify the main theme and subthemes under each theme.

Other issues that requires the authors attention has been highlighted in the manuscript

6. PLOS authors have the option to publish the peer review history of their article (what does this mean? ). If published, this will include your full peer review and any attached files.

**Do you want your identity to be public for this peer review?** For information about this choice, including consent withdrawal, please see our Privacy Policy .

Reviewer #1: No

---

## [Author Response · Author response to Decision Letter 1]

9 Apr 2024

PLEASE SEE SEPERATE FILE ATTACHED WITH THIS SUBMISSION

Title: Contribution of health system governance on delivery of primary health care services towards universal health coverage: a scoping review

Manuscript ID: PONE-D-23-27336

Point by point to the editor's and reviewers’ comments

The authors’ team would like to thank you for your insightful and constructive feedback on our manuscript. We have revised it as suggested. In this document, we have responded point-by-point to your comments and clarified the concerns where necessary.

Additional Editor Comments:

Additional Editor Comments: Response

Title: The title needs to be reviewed. “Governance of primary health care system”. What is the gap in the primary health care system? Are you talking about the influence of governance? Are you talking about the impact of healthcare governance on primary health system? The title needs to be amended. The title and objective must be linked. “In the objective, you have “explores function” and “UHC”. The title should have UHC.

Kindly refer to your research question to formulate the title “Line 115” “We identified the research question focusing on the contribution of HSG to delivery of PHC services”

We have revised the title to include universal health coverage. Our focus was to synthesize the contribution of governance to the delivery of PHC services towards UHC. The manuscript's focus has been revised to include a broader focus on the contribution of HSG in the delivery of PHC services towards UHC.

Introduction

1. The first paragraph of introduction is having some repetitions: Line52 and Line55:

Line52: Health system governance (HSG) is the rules (formal and informal) for collective action

Line55: Additionally, HSG is “as the collective actions…”

Line 59-63: This paragraph is not clear. Kindly break the paragraph in to pieces to make it easy to understand for the reader We have revised these as suggested. Repetitions were removed, and the paragraph is broken and made clearer.

There are inconsistencies between the aim given in abstract and the one given in the introduction.

Line25: In the abstract the aim of the study: “This scoping review explores the function of HSG in the context of PHC setting towards UHC”.

Line102: The aim of the study; “This study aimed to synthesize HSG interventions' contribution to delivering and utilizing PHC services” These were revised and made consistent.

We focused on HSG's contribution to delivering PHC services for UHC.

Page64-82: this paragraph enumerates frameworks: when I read these frameworks, I would expect to see the frameworks utilized in the document but it is not the case. Instead, kindly discuss at least 3 studies on health systems governance of primary healthcare in the introduction and let the 3 studies support the discussion and universal health coverage! We thank the editor for this important insight. As suggested in the introduction and discussion section, we have included studies.

Introduction should also briefly discuss Health systems blocks because these must comeback in the discussion. We revised it as suggested. We included WHO BB in both the introduction and discussion sections.

2. Methodology

This framework of Arksey and O'Malley should be briefly described

We briefly described the framework and followed the steps suggested by the framework.

Line 116: We conceptualized three concepts: role of HSG to deliver health service to achieve UHC; Where are the 3 concepts???? We mentioned these three concepts- HSG, PHC and UHC

Identifying relevant studies

1. What guided the choice of the 6 databases? I have a concern on why you have chosen “PsychInfo” as one of the databases to search from. PsychInfo is a database of abstracts and articles in the field of psychology and psychiatry. Why did you choose this database? This needs a strong justification. 2. For each database, provide the search strategy including the search terms. Provide also the search strategy for google scholar engine.

We described why these databases were included in the search, including using GS. Multiple databases were used to capture important and relevant studies on the topic.

Study selection

It is not clear how you identified relevant studies. Kindly provide clearly inclusion and exclusion criteria

We included the inclusion and exclusion criteria.

Summarizing and reporting

It is confusing how you summarized and reported the results. Kindly elaborate how you did a synthesis in details, how you grouped the papers according to the groups they belong

We mentioned how data were analysed, synthesised and reported in the results section.

3. Results

PRISMA: “Excluded as did not match purpose of the review”. This statement is not simply enough, kindly elaborate for each paper the reason of exclusion.

Line183: “In the final review, we included 64 studies”, yet in the PRISMA, the number of papers are 74.

Descriptive characteristics of included studies We corrected this in the PRISMA and revised it accordingly.

Kindly describe all papers identified by country, type of paper, author and year with citation of each paper Revised accordingly, Table S4.

Move Table1 to supplementary documents Corrected as suggested and moved in the supplementary file.

Line 202: Fig2 presents the summary of HSG interventions in the PHC context

Is this Fig2 part of your results? In which paper did you extract this fig2? If the information of figure2 is not among the results, kindly put it in the introduction or literature review Figure 2 is removed as all supplementary files table S3 table S4.

Line 164: You said that to summarize the findings you used WHO framework on governance and financing: Looking in to the summary, they are many subtitles which are not part of WHO framework such multi-sectoral engagement (line 438), community engagement (line 459),…. These subtitles are themes under each component of the WHO framework. The WHO framework is a guiding framework in which related themes fit in each component and are explained by synthesising the findings from included studies.

Line 570: Our purpose of the review was to synthesize evidence rather than grade the evidence, however, the results are a simple summary of what was reported in the studies, on the contrary, the paper should synthesize the data as opposed to summarize. Results should have a synthesis and not a summary.

We agree with the editor. We synthesized the study’s findings under each component of the framework.

In the aim of the study and methodology, Universal health coverage appeared both in aim and search but does not come out among findings. What happened in the findings? Is it that no studies reported universal coverage? The UHC was reported using the available data from the included studies.

4. Discussion: Discuss how performing HSG (countries) are doing, what special strategies they have in terms of HSG which are making them performing and their strengths and weaknesses. Discuss countries in pools of LMICs and HICs, what are differences We used a similar structure to the findings section. Interpretations are based on the key findings of each component of the framework. We interpreted our findings by referring to various LMICs and HICs to support the interpretation.

Conclusion

The conclusion is not supporting the findings at all. Kindly relate your findings to the presented results We thank the editor, and we revised the conclusion accordingly.

Reviewer 1

The authors need to address inconsistencies in the document, In the abstract, the authors write that they used 64 studies in the review, but in the main document there are 74 studies.

We corrected this mistake.

In addition, the purpose of this scoping review is to explore the function of HSG in the context of PHC settings towards UHC but in the methodology, the research questions focused on contributions of HSG to the delivery of PHC services. The purpose fails to capture UHC which is consistently positioned as very important in the topic of HSG and PHC.

We agreed with the reviewer that we primarily focus on the contribution of UHC in the delivery of PHC, and effective delivery of PHC contributes to the UHC (directly and indirectly). We have reported and interpreted.

It is not clear what was done to the broad research questions, in addition it is not clear which is the broad research question and which is the final reviewed research question. See Lines 119-120

Our main focus was to understand the contribution of HSG in delivery of PHC services to UHC.

Why were the six databases selected, a justification for this choice is essential.

We provided a justification for why we used six databases.

Line 116: The author should discuss the three concepts referred to here in this line. We mentioned these terms

I suggest that authors avoid the use of disagreements by the reviewers on varying issues e.g Line 146…

We corrected it as suggested.

In Line 192, it is not clear how many of these studies were from, El Salvador, Ukraine, Egypt, Indonesia, and Kenya

We specified these studies.

In line 499, digital tools cannot form policies,

In lines 285-line 286 (who is empowered?)

We made clarification.

In lines 142-143. Were there specific interventions the authors were referring to?

Line 87-98 the sentence is incomplete. We revised it and made it clear.

---

## [Decision Letter · Decision Letter 1]

29 Sep 2024

PONE-D-23-27336R1Contribution of health system governance on delivery of primary health care services towards universal health coverage: a scoping reviewPLOS ONE

Dear Dr. Khatri,

Thank you for submitting your manuscript to PLOS ONE. After careful consideration, we feel that it has merit but does not fully meet PLOS ONE’s publication criteria as it currently stands. Therefore, we invite you to submit a revised version of the manuscript that addresses the points raised during the review process.

We look forward to receiving your revised manuscript.

Kind regards,

Sherlyn Villate

Masoud Behzadifar

Academic Editor

Reviewers' comments:

Reviewer's Responses to Questions

**Comments to the Author**

1. If the authors have adequately addressed your comments raised in a previous round of review and you feel that this manuscript is now acceptable for publication, you may indicate that here to bypass the “Comments to the Author” section, enter your conflict of interest statement in the “Confidential to Editor” section, and submit your "Accept" recommendation.

Reviewer #1: All comments have been addressed

Reviewer #2: (No Response)

2. Is the manuscript technically sound, and do the data support the conclusions?

Reviewer #1: Yes

Reviewer #2: Partly

3. Has the statistical analysis been performed appropriately and rigorously? 

Reviewer #1: N/A

Reviewer #2: No

4. Have the authors made all data underlying the findings in their manuscript fully available?

Reviewer #1: Yes

Reviewer #2: Yes

5. Is the manuscript presented in an intelligible fashion and written in standard English?

Reviewer #1: Yes

Reviewer #2: No

6. Review Comments to the Author

Reviewer #1: I am satisfied that the authors have addressed the raised comments. Despite the large number of comments, the authors addressed them well.

Reviewer #2: The introduction is too long and not well structured. The flow is not coming out clearly and it is difficult for the reader to follow. The topic is broad and there is a need to narrow down on the topic. Kindly narrow down and introduce the topic clearly. i.e; How HSG is very different worldwide. High income countries have a different HSG compared to Low and Middle Income Countries. Kindly make distinction between the two and briefly describe the differences between the two by focusing on one geographical area. This study used the “ Arksey and O’ Malley’s scoping review framework” methodology. Kindly describe briefly how the framework supports your methodology. The authors have given so many framework examples but are not focusing on the main framework used. Identify only 3 previous studies on the topic and briefly discuss them in the introduction. What is the research question of this study? Previous review comments on introduction have been addressed partially.

Methodology

Line137-138: “We identified the research question focusing on the contribution of HSG to the delivery of PHC services towards UHC”. Where is the research question??

For methodology, I have a serious problem with the search strategy. The search strategy is not clear including search terms and eligibility criteria, making it impossible to reproduce

Kindly Make two additional headings in methodology, one for “Data sources and Search strategy” and another one “inclusion and exclusion criteria”

It is also not clear how the study selection was conducted and how the synthesis of results was done. Preview comments on methodology have not been addressed

Results

There is a need to clarify step by step how the screening of the results was done by explaining each step how you excluded papers and reasons till 74 included papers.

The descriptive characteristics of included studies is not clear. For each paper included, attach it to a country. For example 3 papers from china, 1 paper from South Africa, …The results of this study are reported in form of summary instead of synthesis. This results provide a concise representation of main points and key findings from the literature reviewed. The study is condensing the information to highlight core aspects instead, the reporting should focus on synthesis by integrating information from different studies to generate insights, theories and understandings that goes beyond the individual contributions of papers.

It is very difficult to comment on discussion and conclusion if the methodology and results are not well done.

7. PLOS authors have the option to publish the peer review history of their article (what does this mean? ). If published, this will include your full peer review and any attached files.

**Do you want your identity to be public for this peer review?** For information about this choice, including consent withdrawal, please see our Privacy Policy .

Reviewer #1: **Yes: ** Dr Kezia Njoroge, Senior lecturer in Public health at Liverpool John Moores University. I give consent to have my full names used in the published peer review

Reviewer #2: No

---

## [Author Response · Author response to Decision Letter 2]

17 Oct 2024

Point by point to the editor's and reviewers’ comments

Manuscript ID: PONE-D-23-27336R1

The authors’ team would like to thank you for your insightful and constructive feedback on our manuscript. We have revised it as suggested. In this document, we have responded point-by-point to your comments and clarified our concerns where necessary.

Reviewer #1(Dr Kezia Njoroge, Senior lecturer in Public health at Liverpool John Moores University)

Reviewer’s comment: I am satisfied that the authors have addressed the raised comments. Despite the large number of comments, the authors addressed them well.

Authors’ response: The authors team would like to thank you for your time and input on our manuscript. Your comments were so insightful and constructive. We incorporated them in the revision and realise that our manuscript improved significantly.

Reviewer #2

Introduction

Reviewer’s comment: The introduction is too long and not well structured. The flow is not coming out clearly and it is difficult for the reader to follow. The topic is broad and there is a need to narrow down on the topic. Kindly narrow down and introduce the topic clearly. i.e; How HSG is very different worldwide. High income countries have a different HSG compared to Low and Middle Income Countries. Kindly make distinction between the two and briefly describe the differences between the two by focusing on one geographical area.

Authors’ response: Thank you, reviewer, for this comment. We have revised this section. Current framing of the introduction section is the concept of health system governance, and its importance, followed by the framework of the framework in relation to primary health care and universal health coverage. Introduction of health system governance of high and low-income counties. The role of health system governance and building blocks, UHC and PHC, was introduced. Finally, the introduction section provides the rationale for the review, study objective and potential implications of the study. We believe that the current framing of the introduction section is more structured and reader-friendly.

Reviewer’s comment: This study used the “ Arksey and O’ Malley’s scoping review framework” methodology. Kindly describe briefly how the framework supports your methodology. The authors have given so many framework examples but are not focusing on the main framework used. Identify only 3 previous studies on the topic and briefly discuss them in the introduction. What is the research question of this study? Previous review comments on introduction have been addressed partially.

Authors’ response: Again, thank you for this feedback. We removed the description of the other framework. We have focused on HSG framework only. We used “Arksey and O’ Malley’s scoping review framework” to guide our methods of scoping review, while the HSG framework is used to present our findings. In short, the former framework guides the research process while later helping to organize the findings. Study research question is included in the last paragraph of the introduction section.

Methodology

Reviewer’s comment: Line137-138: “We identified the research question focusing on the contribution of HSG to the delivery of PHC services towards UHC”. Where is the research question??For methodology, I have a serious problem with the search strategy. The search strategy is not clear including search terms and eligibility criteria, making it impossible to reproduce. Kindly Make two additional headings in methodology, one for “Data sources and Search strategy” and another one “inclusion and exclusion criteria”

It is also not clear how the study selection was conducted and how the synthesis of results was done. Preview comments on methodology have not been addressed

Authors’ response: Thank you for the feedback. We have included the research question in the last paragraph of the introduction section (what are the contribution of HSG (successes and challenges) in the production and delivery of primary health care services towards UHC?). Additionally, we have added two subheadings, as suggested in the methods section.

Results

There is a need to clarify step by step how the screening of the results was done by explaining each step how you excluded papers and reasons till 74 included papers.

The descriptive characteristics of included studies is not clear. For each paper included, attach it to a country. For example 3 papers from china, 1 paper from South Africa, …The results of this study are reported in form of summary instead of synthesis. This results provide a concise representation of main points and key findings from the literature reviewed. The study is condensing the information to highlight core aspects instead, the reporting should focus on synthesis by integrating information from different studies to generate insights, theories and understandings that goes beyond the individual contributions of papers.

Authors’ response: Thank you for the feedback. We have revised it accordingly. The findings are presented based on our methods (framework-guided thematic analysis of the contents derived from studies included in the review). This analysis approach is different from what the reviewer suggested. We believe the current structure and presentation findings are useful for the readers and reader-friendly.

It is very difficult to comment on discussion and conclusion if the methodology and results are not well done.

Authors’ response: We conducted this scoping review by following scoping review guidelines and analysing data using a framework guided by a thematic analysis approach. We presented findings guided by the methods (for instance, narrative explanation of themes under each component of the framework) of past published research. We framed the discussion section based on the findings and concluded based on the discussion and objective of the study. Thus, the content and organization of the discussion section are coherent with the methods and findings.

We thank reviewers for the very positive and constructive feedback on our work, and we are so grateful to the reviewers and editor.

---

## [Decision Letter · Decision Letter 2]

14 Jan 2025

Contribution of Health System Governance in Delivering Primary Health Care Services for Universal Health Coverage: A Scoping Review

PONE-D-23-27336R2

Dear Dr. Khatri,

We’re pleased to inform you that your manuscript has been judged scientifically suitable for publication and will be formally accepted for publication once it meets all outstanding technical requirements.

Kind regards,

Masoud Behzadifar

Academic Editor

PLOS ONE

Additional Editor Comments (optional):

Reviewers' comments:

Reviewer's Responses to Questions

**Comments to the Author**

1. If the authors have adequately addressed your comments raised in a previous round of review and you feel that this manuscript is now acceptable for publication, you may indicate that here to bypass the “Comments to the Author” section, enter your conflict of interest statement in the “Confidential to Editor” section, and submit your "Accept" recommendation.

Reviewer #3: All comments have been addressed

2. Is the manuscript technically sound, and do the data support the conclusions?

Reviewer #3: Yes

3. Has the statistical analysis been performed appropriately and rigorously? 

Reviewer #3: Yes

4. Have the authors made all data underlying the findings in their manuscript fully available?

Reviewer #3: Yes

5. Is the manuscript presented in an intelligible fashion and written in standard English?

Reviewer #3: Yes

6. Review Comments to the Author

Reviewer #3: Respected Editor,

I have evaluated the revised manuscript positively and recommend it for publication.

7. PLOS authors have the option to publish the peer review history of their article (what does this mean? ). If published, this will include your full peer review and any attached files.

**Do you want your identity to be public for this peer review?** For information about this choice, including consent withdrawal, please see our Privacy Policy .

Reviewer #3: No

---

## [Editor Report · Acceptance letter]

PONE-D-23-27336R2

PLOS ONE

Dear Dr. Khatri,

I'm pleased to inform you that your manuscript has been deemed suitable for publication in PLOS ONE. Congratulations! Your manuscript is now being handed over to our production team.

Kind regards,

on behalf of

Dr. Masoud Behzadifar

Academic Editor

PLOS ONE